# Is Dutasteride a Therapeutic Alternative for Amyotrophic Lateral Sclerosis?

**DOI:** 10.3390/biomedicines10092084

**Published:** 2022-08-25

**Authors:** Belén Proaño, Julia Casani-Cubel, María Benlloch, Ana Rodriguez-Mateos, Esther Navarro-Illana, Jose María Lajara-Romance, Jose Enrique de la Rubia Ortí

**Affiliations:** 1Doctoral Degree School, Health Sciences, Catholic University of Valencia San Vicente Mártir, 46001 Valencia, Spain; 2School of Medicine and Health Sciences, Catholic University San Vicente Mártir, 46001 Valencia, Spain; 3Department Nursing, Catholic University San Vicente Mártir, 46001 Valencia, Spain; 4Department of Nutritional Sciences, King’s College London, Franklin Wilkins Building, London SE1 9NH, UK; 5Multimedia Department, Catholic University of Valencia San Vicente Mártir, 46001 Valencia, Spain

**Keywords:** amyotrophic lateral sclerosis, dutasteride, neuroprotection, oxidative stress, inflammation

## Abstract

Amyotrophic lateral sclerosis (ALS) is a neurodegenerative disease that is characterized by the loss of upper and lower motor neurons (MNs) in the cerebral cortex, brainstem and spinal cord, with consequent weakness, atrophy and the progressive paralysis of all muscles. There is currently no medical cure, and riluzole and edaravone are the only two known approved drugs for treating this condition. However, they have limited efficacy, and hence there is a need to find new molecules. Dutasteride, a dual inhibitor of type 1 and type 2 5α-reductase (5AR) enzymes, the therapeutic purposes of which, to date, are the treatment of benign prostatic hyperplasia and androgenic alopecia, shows great anti-ALS properties by the molecular-topology methodology. Based on this evidence, this review aims to assess the effects of dutasteride on testosterone (T), progesterone (PROG) and 17β-estradiol (17BE) as a therapeutic alternative for the clinical improvement of ALS, based on the hormonal, metabolic and molecular pathways related to the pathogenesis of the disease. According to the evidence found, dutasteride shows great neuroprotective, antioxidant and anti-inflammatory effects. It also appears effective against glutamate toxicity, and it is capable of restoring altered dopamine activity (DA). These effects are achieved both directly and through steroid hormones. Therefore, dutasteride seems to be a promising molecule for the treatment of ALS, although clinical studies are required for confirmation.

## 1. Introduction

Amyotrophic lateral sclerosis (ALS) is a neurodegenerative disease that is characterized by the loss of upper and lower motor neurons (MNs) in the motor cortex, brainstem and spinal cord, with consequent weakness, atrophy and the progressive paralysis of all muscles. The final consequence is patient death due to respiratory failure from 2 to 4 years from the diagnosis of the disease [1]. Worldwide, the incidence ranges from 0.42 to 2.76 per 100,000 person-years, and the prevalence ranges from 1.57 to 9.62 per people, with a higher proportion of men for both variables, reaching a peak between 60 and 70 years of age [2].

Despite being a disease with a documented history of 160 years, and more than 200 clinical trials, a cure has yet to be discovered. Only the drug Riluzole^®^, a glutamate antagonist, is approved worldwide, increasing the average survival in patients with bulbar ALS by 3 months mainly, and reducing the deterioration of muscle strength [3]. In addition, the antioxidant Edaravone^®^ is approved in the United States and Japan, and it slows down the progress of the disease in its initial stages [4].

Therefore, there is clearly a need to identify new treatment options for ALS, and computational methods based on receptors and ligands are known to be efficient tools in terms of time and resources. To date, this technology has enabled the identification of new treatments for diseases of the central nervous system (CNS), such as Alzheimer’s disease (AD) [5], cancer [6,7] and even SARS-CoV-2 [8]. Specifically for ALS, up to 10 compounds, which could be potential candidates for the treatment of this disease, were identified by molecular topology. One of the them was the drug dutasteride [9], which showed favorable docking scores and stable interactions with the relevant amino acids of the TDP-43 protein, the mutations of which are related to the development of ALS (see Section 2.3). Dutasteride is used to treat benign prostatic hyperplasia and androgenic alopecia [10]. It delays the progression of prostate cancer and the start of aggressive therapies [11], and it has even contributed to the recovery of male patients with COVID-19, where being male is linked to a worse prognosis [12].

Dutasteride is a dual inhibitor of type 1 and type 2 5α-reductase (5AR) enzymes. These enzymes catalyze the saturation of the 4,5 double bond of the A ring of Δ4-3 ketosteroids, such as testosterone (T) and progesterone (PROG), into dihydrotestosterone (DHT) and 5α-dihydroprogesterone (DHP), respectively [13]. The modulation of 5AR activity can also indirectly increase the levels of 17β-estradiol (17BE), derived from the aromatization of T [14]. T and PROG, together with their steroid derivatives, have been shown to have neuroprotective effects, and their alteration has been linked to the pathogenic mechanisms of ALS [15]. Therefore, the aim of this review is to assess the activity of dutasteride, and the hormones it boosts, as a therapeutic alternative for the clinical improvement of ALS, based on the hormonal, metabolic and molecular pathways related to the pathogenesis of ALS disease.

## 2. Pathogenesis of Amyotrophic Lateral Sclerosis

### 2.1. The Role of Steroid Hormones

Sex steroids have been shown to influence the etiology and pathophysiology of ALS. Epidemiologically, there are differences in the age of onset of the first symptoms and the progression of the disease depending on sex. In a clinical analysis carried out over 5 years, a mean age of between 60 and 63 years was reported for the diagnosis of the disease in men and women, respectively, with a greater initial involvement of the bulbar region in women, and 11% less in men. Moreover, although men are more like to develop ALS, for people over 50 years of age, the risk was the same, with a decrease in the male:female ratio from 2.5:1 to 1:1 after that age [16,17,18,19]. Steroid hormones play a modulatory role in the development, differentiation and function of the CNS from intrauterine life. Free testosterone (FT) crosses the blood–brain barrier (BBB) and exerts its direct functions in the CNS by activating androgen receptors (ARs), or, as described, by being a substrate for the transformation of the other steroid hormones, such as DHT or 17BE, by means of the 5AR or aromatase enzymes, respectively [20]. Similarly, PROG crosses the BBB, transforming into DHP through 5AR, and both are able to interact with the PROG receptor, which is mainly involved in myelin synthesis and oligodendrocyte maturation [21]. This is why the link between the alteration of these hormone levels and the development of neurodegenerative diseases has been extensively explored, showing that their decrease as the patient ages, and especially T [22], is a probable cause for developing Alzheimer’s disease (AD) [23], multiple sclerosis (MS) [22], Parkinson’s disease (PD) [24] and ALS [25].

Specifically, in ALS, an alteration of the sex hormones has been confirmed, although the role of androgens or estrogens has not been established with certainty [26]. Lower estrogen and PROG levels have been found compared with healthy controls [27], and elevated PROG levels correlate with favorable prognostic factors. The sex bias in the ALS incidence, which disappears after the age of 55 years, would also suggest that higher levels of circulating estrogen in premenopausal women may be protective [28]. However, androgen activity is more controversial. It has been described that T mediates MN survival, the dendrite length [29] and axonal regeneration after injury [30], in addition to being antiapoptotic [31], which are processes related to the pathophysiology of ALS [32]. However, high circulating prenatal T levels have also been associated with an increased risk of ALS [33]. In fact, a higher risk of suffering from ALS has been suggested in elite athletes in contact sports, which could be mediated by exposure to high levels of T during intrauterine life [34], and hence, their lower digit ratio (2D:4D) [33]. Moreover, in adult life, abnormally high levels of T are related to a poorer respiratory capacity [35], and, in this context, high blood levels of total T and FT are linked to a significant monthly decrease in the forced vital capacity (FVC) in ALS patients. However, it has also been seen that this influence could be compensated for by raising the PROG levels and, therefore, the PROG/TL ratio in plasma [36]. This effect of PROG could be due to the location of the sex steroid receptors on respiratory MNs [35], and, in this context, PROG stimulates the ventilation per minute at rest by acting on the central respiratory regions (nucleus tractus solitarius, medulla oblongata and structures of the hypothalamic nuclei) [37,38,39]. Moreover, with these altered FT levels, the hypothesis of “testosterone resistance in the blood-brain barrier” should also be considered [40]. This hypothesis proposes that people with a predisposition for ALS have a BBB dysfunction that prevents the passage of T to the CNS, and so they develop a compensatory mechanism with higher serum TF levels, which maintain T concentrations within the CNS and avoid DHT [40] or 17BE deficiencies [41]. When the compensatory mechanism fails, probably due to age, the disease sets in, coinciding with low DHT levels in the CSF. Therefore, ALS patients around 50 years of age presented elevated T and FT levels in serum [36], and older patients with a mean age of 63 years showed lower FT levels in both sexes [42]. These apparent differences would confirm the theory of “resistance to testosterone in the blood-brain barrier” because the age of patients with low T levels is approximately 10 years older than in the other studies, which could coincide with the time the compensatory mechanism fails due to the effect of age.

Another important aspect would be the involvement of sex hormones in the regulation of the cognitive function, in the neuroimmune responses and, especially, in the altered motor control in ALS due to their action on the signaling pathways of catecholaminergic neurons and cholinergic systems [22]. In this sense, significant alterations were found in several metabolic pathways in TDP-43G298S mutant mice, such as the positive regulation of the metabolites of the tricarboxylic acid cycle, and defects in the levels of neurotransmitters, such as dopamine (DA). As a matter of fact, an increase in DA is related, in turn, not only to improvements in the locomotor function in vivo in the ALS Drosophila model for a TDP-43 proteinopathy [43], but also to a better cognitive status in ALS patients, given the significant decrease in the DA receptors in some areas outside the striatum [44].

### 2.2. Mitochondrial Damage and Oxidative Stress

The malfunction of the mitochondria as an organelle that supplies cellular energy is closely related to MN death [45]. In this regard, the “lactate dyscrasia” theory suggests that mitochondrial respiration impairment causes toxicity and neuronal degeneration due to lactate accumulation in the neuromuscular junction, such that the neuron loss would resemble a dying-back process [46]. In fact, lactic acid is among the nine altered metabolites (along with maltose, glyceric acid, beta-alanine, phosphoric acid, glutamic acid, ethanolamine, glycine and 2,4,6-tri-tert-butylbenzenethiol) that help to differentiate ALS patients from healthy controls, and to even differentiate between patients with rapid and slow progression [47]. This is why restoring mitochondrial alterations is closely linked to new therapeutic approaches for ALS, as these alterations are characterized by mitochondrial electron-transport-chain impairment [48], mitochondrial DNA deletions and reduced ATP levels [49], which are present in both patients with familial ALS (fALS) and sporadic ALS (sALS). These mitochondrial dysfunctions have linked oxidative-stress (OS) increases, antioxidant-defense decreases and inflammatory processes with the pathogenesis of ALS [50]. As a consequence, ALS patients have increased levels of OS markers, such as malondialdehyde or 8-hydroxy-2′-deoxyguanosine (8-OHdG), as well as proinflammatory cytokines, such as IL-6 and IL-8, which confirm the systemic alteration of the redox state and the inflammation in these patients [51].

### 2.3. Mutations and Abnormal Protein Aggregation: SOD1, C9ORF72, TDP-43, FUS, VCP

More and more evidence is emerging on the mutations responsible for the aggregation and misfolding of the proteins involved in cellular processes and their influence on many progressive neurodegenerative disorders, such as AD, PD and ALS itself [26]. The most frequent mutations in fALS and sALS are those that affect the superoxide dismutase-1 genes (SOD1), present in 1–3% of sALS cases, and intronic expansions in the ORF72 region of chromosome 9 (C9ORF72), found in 5–10% of sALS cases [52]. 

In addition, much of the animal-model research focuses on mutations in the TARDBP transactive response DNA-binding protein gene (TDP-43) associated with approximately 5% of fALS cases [53], sarcoma fusion protein (fused in sarcoma (FUS)) and valosin-containing protein (VCP), due to its link to fALS and diseases of a similar nature [52,54].

SOD1: Alterations in the expression of this enzyme are linked to the deregulation of mitochondrial-matrix homeostasis. The accumulation of mutant SOD1 proteins in ALS could be the cause of tissue damage and OS. For example, in the SOD1G93A mouse model, mutated-protein accumulation in the mitochondria and cytosol causes impaired energy metabolism and OS in the lungs, with the lower activity of antioxidant enzymes, such as glutathione reductase or catalase [55]. In addition to the loss of function of antioxidant enzymes, mutations in SOD1 reduce the protein-folding stability through the disruption of metal binding and/or disulfide-bond formation, resulting in misfolding, aggregation and ultimately cell toxicity [56].

C9ORF72 mutations: The hexanucleotide repeat expansion (GGGGCC (G 4 C 2)) (HRE), located in the first intron of the C9ORF72 gene, has been related to the pathogenesis of ALS, as it has been seen that these expansions generate a long RNA that sequesters RNA-binding proteins, promoting the formation of other protein inclusions, such as TDP-43 [57]. In Europe, up to 50% of fALS patients, and approximately 5% of sALS patients, have these expansions [58].

TDP-43: This is a protein that is involved in the regulation of various aspects of RNA processing, including RNA splicing, transport, the RNA stability and the production of microRNAs, and it has the ability to shuttle between the nucleus and cytoplasm [59]. Under pathological conditions, post-translational structural modifications of TDP-43 cause mislocalization, with abnormal accumulation in the neuron cytoplasm, which is accompanied by proteolytic cleavage and abnormal C-terminal fragments. In particular, alterations in the RNA recognition domain (RRM1) of TDP-43, induced by OS, can promote its aggregation and incorrect localization in the cytoplasm [60]. These abnormal protein aggregates are the most representative in ALS disease [54,61,62].

FUS: Similar to TDP-43, FUS is an RNA-binding protein that has the ability to modify RNA metabolism, with a preferentially nuclear localization. Mutations in the nuclear domain of FUS have been related to impaired nuclear transport and poor cytoplasmic localization, and mainly in juvenile ALS [63]. Jensen et al. demonstrated, in an intraspinal-cord-injection model, that restricted expression of mtFUS in astrocytes is sufficient to induce the death of spinal MNs, which would produce motor deficits mediated by an increase in TNFα. Furthermore, they showed that TNFα is a key toxic molecule, as the expression of mtFUS in TNFα knockout animals does not induce pathogenic change. Consequently, in mtFUS-transduced animals, the administration of TNFα-neutralizing antibodies prevents neurodegeneration and motor dysfunction [64]. Increased FUS levels in astrocytes also lead to an increased proinflammatory microglial response [65].

VCP Protein: This protein is related to DNA chain repair when damaged and is essential for the maturation of ubiquitin-containing autophagosomes. The mutant variant is toxic, partially mediated by its effect on the TDP-43 protein. VCP mutations may represent between 1 and 2% of fALS, and they show the direct involvement of the protein ubiquitination/degradation-pathway impairment in MN degeneration [66].

### 2.4. Neuroinflammation: Activation of Classical NF-κB Pathway

Neuroinflammation, which is mainly characterized by the presence of reactive astrocytes and microglia, the moderate infiltration of peripheral immune cells and high levels of inflammatory mediators, affects the motor regions of the CNS in fALS and sALS [67]. Microglia are the resident immune cells of the CNS [68], and, if an insult is not resolved, they will remain reactive, which means that astrocytes and oligodendrocytes continue to be recruited, and neurodegeneration and the inflammatory process progress [69]. The role of nuclear factor kappa B (NF-κB) is also noteworthy. It acts as an inflammation regulator and is clearly increased in the spinal cords of ALS patients and SOD1G93A mice. Thus, it has been observed that the deletion of NF-κB signaling in microglia rescued MNs from microglial-mediated death in vitro, and prolonged survival in SOD1G93A mice [70]. Several studies have shown that the inhibition of the Keap1–Nrf2 interaction can, on the one hand, eliminate reactive oxygen species (ROS) or inhibit the transcription of proinflammatory cytokine genes [71,72,73], and, on the other hand, inhibit NF-κB activation, suppressing inflammation [74].

In addition, endoplasmic-reticulum (ER) stress, which occurs when the ER–mitochondria calcium cycle is disrupted and misfolded proteins accumulate in the ER, is also involved in the pathogenesis of ALS. To cope with ER stress, cells activate the unfolded protein response (UPR). The evidence in non-neuronal cell models suggests that there is a strong interaction between the UPR and NF-κB pathway, showing a link between these two important pathogenic mechanisms of ALS (ER stress and NF-κB signaling) in MNs [75].

## 3. Therapeutic Effect of Dutasteride

### 3.1. Role of Dutasteride in the Activity of Steroid Hormones

Steroid hormones have been linked to neuroprotection because they participate in anti-inflammatory processes, are antioxidant, stabilize the BBB and reduce the accumulation of β-amyloid protein (AB) and excitotoxicity caused by glutamate excess [76]. Dutasteride administration in patients with benign prostatic hyperplasia diminishes the prostate size due to the DHT decrease and T increase [77].

In ALS, this increase in the serum T levels could help maintain the plasma FT levels, which can readily be aromatized in the CNS after crossing the BBB [78]. Similarly, the inhibition of the PROG transformation into DHP could result in a free PROG increase so that its neuroprotective effects can be seen.

PROG in the CNS stimulates myelin production mainly in animal models of neurological disorders [79]. Clinically, it is also considered to be a molecule with therapeutic potential because, due to its low molecular weight and fat-soluble nature, it diffuses easily into nervous tissue, in addition to interacting with multiple targets [80]. For example, patients with traumatic acute brain injury who took PROG had lower 30-day mortality and a better prognosis than the placebo group in the ProTEC trial [81]. Low levels of PROG (in men and women) have also been linked to bulbar ALS, a shorter time from symptom onset to diagnosis and a shorter survival time from diagnosis [79]. In a case–control study in menopausal women with ALS, longer reproductive periods (from menarche to menopause) were associated with a lower risk of developing the disease and longer survival, suggesting endogenous-estrogen mediation [82].

It has been suggested that both PROG and 17BE have a neuroprotective effect, mainly due to the lower proportion of women suffering from the disease. In addition, diagnosis is more frequent after menopause, and very few cases of pregnant women with ALS or pregnancies after the onset of the disease have been reported. On the rare occasion that the latter has occurred, pregnancy seems to have attenuated the symptoms of the pathology [83]. 17BE and T can also inhibit the activation of microglia cells, playing an anti-inflammatory role [84,85]. Furthermore, this anti-inflammatory effect seems to be enhanced with the combination of PROG administration, probably through the inhibition of the signaling pathways that lead to the activation of proinflammatory genes [86], with neuromotor benefits observed using a well-validated mouse model of lipopolysaccharide (LPS)-induced intrauterine inflammation (IUI) [87].

With regard to T, it restored the BBB selective permeability and tight-junction integrity and almost completely suppressed proinflammatory cytokine production, such as TNFα [88]. Furthermore, as already pointed out, DA has been linked to the pathogenesis of ALS and D2 dopamine receptor agonism, which modulates neuronal excitability and may increase MN survival [89]. In this sense, the administration of dutasteride increases the T in the brain in 1-methyl-4-phenyl-1,2,3,6-tetrahydropyridine (MPTP)-lesioned mice, which is related to a clearly beneficial effect in the DA activity due to the reversion of the striatal DA levels [41].

Moreover, insofar as the increase in FT has been related to the loss of the respiratory capacity in ALS, it should be remembered that dutasteride not only increases the T levels, but also the PROG levels. More precisely, Littim (2017) showed that the peripheral and brain levels of both hormones remain high after administering dutasteride [90].

All these mechanisms, summarized in Figure 1, could have a direct clinical impact in ALS patients, and specifically, regarding the possible benefits of T, it should be noted that androgenic therapies have been especially effective in the treatment of neurodegenerative diseases [23] to improve muscle mass and restore the function of the neuromuscular junction [91]. Clinically, muscle atrophy and a loss of strength are responsible for the physical disability in ALS. Androgen therapy stimulates MN recovery, leading to improved neuromuscular function [92] and increased lean mass [93]. These data seem to be supported by the recent work of Yu Jin Kim and colleagues, which showed that long-term hormone therapy (HT) is associated with a greater reduction in the risk of neurodegenerative diseases, and which highlighted the need for precision hormone therapy, as there are multiple factors that can influence its efficacy and safety (such as hormonal fluctuations during peri- and postmenopause associated with changes in the neuroimmune and genetic systems) [94].

### 3.2. Regulatory Mechanisms of Protein Aggregation

The molecular-topology method that identified the therapeutic potential of dutasteride in ALS was based on a discriminant analysis using three computational models that considered the general activity and specific activity in ALS clinical trials and TDP-43 molecular binding due to its strong relationship with the pathogenesis of the disease.

Dutasteride showed binding affinity by hydrogen bond for the analyzed domains 4IUF and 4BS2 [9]. Both domains belong to the RRM family (RNA recognition motif), which consists of RNA-binding proteins involved in the recognition of proteins and post-transcriptional RNA processes [95,96], which can be found in TDP-43 and FUS [97]. Although the interaction sites with RRM reported for dutasteride by Gálvez et al. are a lysine (Lys145) and an aspartic acid (Asp174), the interaction potential is important because RRM1 oxidation has been shown to decrease the TDP-43 solubility and promote aggregate formation [60]. In this context, Liu et al. have proposed a new mechanism of TDP-43 aggregation, which could characterize the formation of large aggregation models with repeated helical and rope-like structures, helping to understand the amyloid-like aggregation phenomena of the TDP-43 protein in ALS [98].

Regarding mutated proteins, mice with SOD1 mutations present specific sex differences in the age of onset and the progression of the disease, which have been related to deficiencies in the expression of androgen receptors (ARs) in the spinal cord [99]. In fact, the same group found that AR depletion was associated with 5AR type 2 depletion [100].

Similarly, VCP mutants are known to bind to different polyglutamine (PolyQ) disease proteins, including AR variants. These abnormal aggregates disrupt the interaction of the double-strand break repair proteins, which, in turn, causes further damage to the DNA double helix. Fujita et al. performed an immunoprecipitation assay to test the interaction between VCP and polyQ disease proteins, finding that VCP binding to AR is dependent on the T concentrations, which suggests that the AR binding of ligands decreases the association with VCP, and thus, the formation of protein aggregates. Therefore, a T increase could also counteract the formation of aggregates dependent on AR binding, as is the case of not only VCP, but also TDP-43 [101]. This, in turn, would counteract C9ORF72 mutations, which are especially involved in the pathophysiology of sALS [57]. Conversely, it would be necessary to consider the possible relationship of these altered proteins with the increased steroid hormones after the administration of dutasteride. However, there are not many studies on this matter, although, interestingly, it has only been described that an increase in 17 β-estradiol enhanced autophagy and suppressed apoptosis to limit MN death in an NSC34 cell-like model that stably expresses the 25 kDa C-terminal fragment of TDP-43 [102]. 17 β-estradiol showed neuroprotector effects in a zebrafish model of C9ORF72-amyotrophic lateral sclerosis [103], which possibly confirms the 17 β-estradiol neuroprotective activity that is also in other protein alterations, such as C9ORF72.

### 3.3. Neuroprotective and Antioxidant Effects of Dutasteride

Neuroinflammation and OS are closely linked in the pathogeneses of neurodegenerative diseases [104]. Infiltrated and glial immune cells are one of the main producers of reactive oxygen species (ROS) and reactive nitrogen species (RNS) in CNS pathologies [105]. Although ROS are not believed to be the cause of ALS, they contribute to the progression of the disease [67].

Dutasteride has shown neuroprotective effects against glutamate toxicity in animal models. In a cell-viability-detection assay on mouse cortical neurons, the aim was to assess which molecules were more effective at counteracting the glutamate toxicity directly related to the pathogenesis of the disease. After testing 146 natural products and 424 FDA-approved drugs to determine their ability to protect neurons against NMDA (N-methyl-D-Aspartate)-induced cell death, dutasteride stood out, together with enalapril and finasteride. By using the in vivo imaging of primary cortical neurons labeled with tetramethylrhodamine ethyl ester, dutasteride was shown to attenuate the NMDA-induced breakdown of the mitochondrial membrane potential [106]. This finding would confirm the previously detected neuroprotective effect of dutasteride; it has been observed that it protects against chemical ischemia and the mitochondrial permeability transition in cultured neurons, which could be due to its role in the modulation of voltage-gated potassium channels [107].

No previous studies have shown the mechanisms and direct effect of dutasteride on the OS reduction in ALS. However, it is possible that the neuroprotective effects of dutasteride are again due to its hormonal products. FT shows great neuroprotective properties, as it has been observed that it improves human neurons and astrocyte survival, acting directly on the mitochondrial membrane, inhibiting the generation of ROS [108,109] and RNS [110] and increasing the sirtuin-1 expression [111]. In cell cultures and male animal models, T stimulates neuronal differentiation, maintaining neuronal plasticity [112], promoting synaptic density [113], increasing the connectivity of hypothalamic neurons [114] and stimulating neurite outgrowth [115]. This neuroprotective effect has been seen in female mice with induced spinal-cord injury, where T produced dendrite-length reduction, weight loss and muscle-fiber atrophy in portions of the quadriceps [2].

However, the neuroprotective effect of PROG should also be added to this T neuroprotective effect [116], which can be particularly seen in the Wobbler mouse model of ALS, and shows the selective loss of MN, astrocytosis and microgliosis in the spinal cord. In this model, PROG decreased the MN vacuolization with the preservation of the mitochondrial respiratory complex I activity, decreased the mitochondrial expression and nitric oxide synthase activity, increased manganese-dependent SOD (MnSOD), stimulated brain-derived neurotrophic factor, increased MN cholinergic phenotype and improved survival, with a concomitant decrease in the cell-death pathways. PROG also showed differential effects on glial cells, including an increased oligodendrocyte density and the downregulation of astrogliosis and microgliosis. These changes are associated with reduced anti-inflammatory markers and increased survival and muscle strength [117].

Finally, it has been seen that MNs express high levels of 5α-reductase enzymes in spinal and bulbar muscular atrophy, which is a disease of a similar nature to ALS. This could imply that alterations in the T conversion are linked to neurodegenerative processes related to MN damage and neuroinflammation [118].

### 3.4. Efficacy of Dutasteride against Neuroinflammation

There is no direct evidence of the effects of dutasteride on inflammation in ALS patients, but it has been shown to decrease the gene-expression levels of the proinflammatory cytokines IL-1b and IL-18 in rats with prostate cancer [119]. Therefore, dutasteride might also have important anti-inflammatory activity. In fact, in vitro, dutasteride significantly reduces the secretion of both IL-6 and TNFα in lipopolysaccharide (LPS)-stimulated BV2 cells and decreases the microglia activation in the brain, hippocampus and plasma in an LPS-induced neuroinflammatory mouse model. This seems to demonstrate that dutasteride effectively suppresses the inflammatory response stimulated by LPS in the peripheral and central nervous systems, and that it could counteract the levels of inflammation markers and oxidation produced by abnormal protein aggregates [120]. In particular, there could be an anti-inflammatory effect with respect to FUS aggregates, the increase of which is proportional to the proinflammatory activity of microglia, mediated mainly by the TNFα increase.

Moreover, dutasteride can inhibit the Keap1–Nrf2 interaction, which seems to represent an important pathway in the development of ALS, by promoting inflammation mediated by an increase in OS, as already described [121].

Dutasteride activity related to the regulatory mechanisms of protein aggregation, neuroprotective and antioxidant effects, and neuroinflammation, is summarized Figure 2.

### 3.5. Possible Side Effects of Dutasteride 

In general, dutasteride appears to be a fairly safe drug, and so its use is recommended. In this respect, a literature review by Hirshburg et al., in 2016 analyzed studies related to adverse events of 5-alpha reductase inhibitors in relation to prostate cancer, psychological effects, their use in women and sexual health.

In large and representative populations, an increase in the incidence of prostate cancer, or an increase in high-grade prostate cancers upon detection, or a variation in the survival rate were not associated with dutasteride. A direct link between the use of 5-alpha reductase inhibitors and depression was not established either. The same revision indicated that there were not many studies on the use of 5-alpha reductase inhibitors in women, but the current known risks to women include birth defects in male fetuses if taken during pregnancy, a decreased libido, headache, gastrointestinal problems, isolated cases of menstrual changes, acne and dizziness. Finally, it was reported that erectile dysfunction, a decreased libido and ejaculation disorders occurred in a fairly low percentage of men [122]. 

In addition, the long-term effect of dutasteride on sexual alterations is interesting as it was associated with a significant increase in impotence, a decreased libido and ejaculation disorders during the first year when compared with a healthy control, but there were no significant differences between the two groups in the second year [123].

## 4. Conclusions

Dutasteride is a possible candidate for the treatment of ALS. This molecule shows neuroprotective effects against glutamate toxicity in animal models, which opens new paths due to the direct implication of the neurotransmitter excitability in ALS. In addition, it also shows anti-inflammatory activity, as it has been observed that it reduces the secretion of both IL-6 and TNFα in vitro and decreases the activation of microglia in the brain. Reducing the TNFα increase could counteract the inflammation and oxidation levels caused by abnormal protein aggregates, which, together with its ability to inhibit the Keap1–Nrf2 interaction, would complete its anti-inflammatory activity. In turn, steroid hormones, increased by the inhibition of the 5AR activity due to dutasteride, are shown to be powerful anti-inflammatory and antioxidant agents, and they are capable of stabilizing the BBB and reducing the excitotoxicity caused by excess glutamate. T, PROG and 17BE can also inhibit microglial activation, thus playing a neuroprotective role, and especially in relation to the altered DA activity in the disease. Regarding the mutations and protein aggregates identified in ALS, the increase in T could also counteract the formation of aggregates of FUS, VCP and TDP-43, which, in turn, are linked to C9ORF72 mutations. All of these processes are directly related to the pathogenesis of ALS, which is why hormone therapy seems to be a good alternative against it.

The activity of dutasteride and its hormonal products could clinically improve ALS. On the basis of the extensive muscle deterioration and atrophy in this type of patient, androgen therapies restore the neuromuscular-junction function and increase the muscle mass, which improve the functional capacity and quality of life. Because there are few studies in this field, the possible benefits of dutasteride administration as a therapeutic alternative for ALS should be studied in depth, and especially its impact on the steroid-hormone levels in serum and CSF, and its relationship with protein aggregates.

## Figures and Tables

**Figure 1 biomedicines-10-02084-f001:**
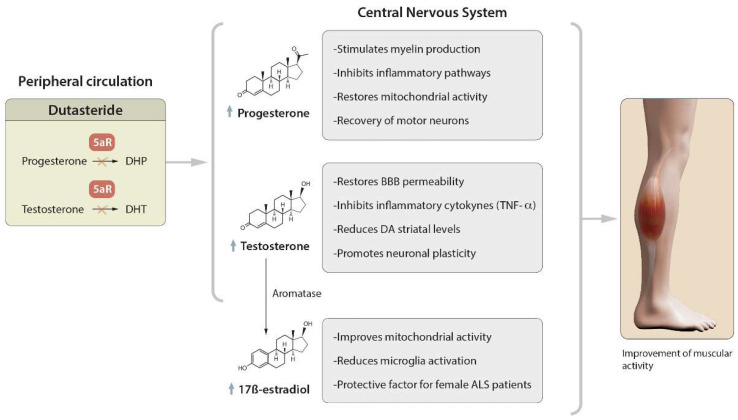
Therapeutic action of dutasteride through steroid hormones. In peripheral circulation, the inhibitory effect of 5AR by dutasteride will reduce the dihydroprogesterone (DHP) and dihydrotestosterone (DHT) levels, with the consequent possible increase (in the progesterone (PROG) and testosterone (T) levels. Both PROG and T will be more available to cross the blood–brain barrier and exert their therapeutic actions in the central nervous system. Moreover, T is substrate for the synthesis of 17 β-estradiol (17BE), which also possesses a neuroprotective role. In sum, these properties should restore the muscular activity in ALS patients. ↑: increase of PROG, T or 17BE blood levels.

**Figure 2 biomedicines-10-02084-f002:**
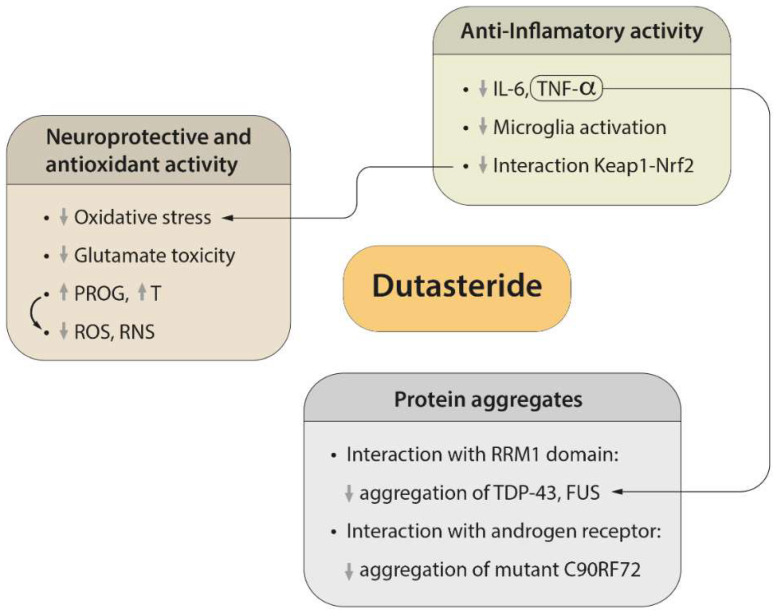
Dutasteride main effects to improve clinical outcome in ALS patients. Dutasteride decreases oxidative stress, including reactive oxygen species (ROS) and reactive nitrogen species (RNS), as well as glutamate toxicity, and increases progesterone (PROG) and testosterone (T) in systemic circulation, which promotes neuroprotection. It also has anti-inflammatory activity that prevents microglia activation and reduces the pro-inflammatory cytokines IL-6 and TNF-α. Moreover, it interacts with Keap1–Nrf2, which, in turn, contributes to the anti-inflammatory effect and oxidative stress. Finally, because of the interaction with the RRM domain, it could reduce the aggregation of TDP-43 and FUS, and because of the interaction with the androgen receptor, it may reduce the aggregation of mutant C9ORF2. ↓: decrease of certain activity (oxidative stress, glutamate toxicity, microglia activation, interaction Keap1-Nrf2, or aggregation of TDP-43, FUS or mutant C90RF72), or PROG, T, IL-6 or TNF-α blood levels, or ROS/RNS decrease. ↑: increase of PROG or T blood levels.

## Data Availability

Not applicable.

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
