# Peer review of "Is Dutasteride a Therapeutic Alternative for Amyotrophic Lateral Sclerosis?"

_biomedicines, 2022, doi:10.3390/biomedicines10092084_

Round 1

Reviewer 1 Report

This review summarizes how dutasteride, a drug normally used to treat benign prostatic hyperplasia and androgenic alopecia, could be a possible drug to treat ALS. The point of the review is interesting, and the authors did a deep analysis of the field. My only concern is that some concepts are hard to follow because they are written in broken English. Before resubmitting the paper, I suggest the authors use editing help from someone with full professional proficiency in English.

Author Response

Thank you for your comment. In response to the reviewer, a full editing has been carried out for better understanding. Due to the many changes, the track changes has been deactivated for that part.

Reviewer 2 Report

The manuscript summarizes the pathogenesis of amyotrophic lateral sclerosis (ALS) as well as the evidence showing therapeutic effect of dutasteride to improve clinical outcome in ALS. The topic is timely and interesting. The manuscript is overall balanced and well-organized.

Issues that need to be addressed:

1.     Advances in understanding the molecular mechanisms of therapeutic potentials of dutasteride in ALS should be mentioned and discussed.

2.     The discussion could be made more interesting by discussing the possible relationships between ALS-associated proteins and steroid hormones in the progression of ALS.

3.     Please summarize possible side effects of dutasteride as a drug.

Author Response

  1. Advances in understanding the molecular mechanisms of therapeutic potentials of dutasteride in ALS should be mentioned and discussed.

We appreciate the comments provided by the reviewer. Regarding the information on the advances in understanding the molecular mechanisms of therapeutic potentials of dutasteride in ALS, to date there are no published studied that analyze these mechanisms in depth. However, in an attempt to improve this part of the article, the information in the text has been completed and discussed, based mainly on its activity against glutamate (page 8), as well as the results obtained from the development of three computational models (on which the Molecular Topology methodology is based), and their activity against oxidative stress and inflammation related to ALS (pages 2, 8 and 9).

  1. The discussion could be made more interesting by discussing the possible relationships between ALS-associated proteins and steroid hormones in the progression of ALS.

It is indeed interesting to check this relationship indicated by the reviewer. Searches have been made that could relate all the altered proteins in ALS (SOD1, C9ORF72, TDP-43, FUS, VCP) with steroid hormones, only finding certain evidence on the role of 17 β-estradiol. This information has been added to the article in section 3.2. Thank you very much.

  1. Please summarize possible side effects of dutasteride as a drug.

Thank you very much for your comment. In response to the reviewer, a section has been added (3.5. Possible side effects of dutasteride) indicating the main adverse effects of the drug.